# The Role of *Serendipita indica* (*Piriformospora indica*) in Improving Plant Resistance to Drought and Salinity Stresses

**DOI:** 10.3390/biology11070952

**Published:** 2022-06-23

**Authors:** Mohammad Reza Boorboori, Hai-Yang Zhang

**Affiliations:** College of Environment and Surveying and Mapping Engineering, Suzhou University, Suzhou 234000, China; boorboori@ahszu.edu.cn

**Keywords:** plants, morphophysiological mechanisms, molecular mechanisms, endophytic fungi, environmental stresses

## Abstract

**Simple Summary:**

Environmental stresses are one of the biggest threats to modern agriculture, and climate change has heightened the risks of these stresses in different parts of the world. Among all the environmental stresses, salinity and drought are a severe threat to arid and semi-arid regions of the world, and for a long time, scientists have been searching for ways to reduce the risk of these stresses. In recent decades, solutions have been developed to reduce the risk of environmental stress on plants by identifying beneficial soil microorganisms. This study was conducted to identify morphophysiological and molecular changes of plants in coexistence with *Serendipita indica* and their impact on drought and salinity stress reduction. The study also has investigated the stressors’ impact on plants and the plants’ mechanisms to cope with them; Furthermore, sharing results with researchers provides a clear path for future research.

**Abstract:**

Plant stress is one of the biggest threats to crops, causing irreparable damage to farmers’ incomes; Therefore, finding suitable, affordable, and practical solutions will help the agricultural economy and prevent the loss of millions of tons of agricultural products. Scientists have taken significant steps toward improving farm productivity in the last few decades by discovering how beneficial soil microorganisms enhance plant resistance to environmental stresses. Among these microorganisms is *Serendipita indica*, which the benefits of coexisting this fungus with plant roots have been extensively explored in recent years. By investigating fungus specification and its effects on plants’ morphological, physiological, and molecular traits, the present study seeks to understand how *Serendipita indica* affects plant resistance to salinity and drought conditions. Furthermore, this study attempts to identify the unknown mechanisms of action of the coexistence of *Serendipita indica* with plants in the face of stress using information from previous studies. Thus, it provides a way for future research to assess the impact of this fungus on tackling environmental stresses and enhancing agricultural productivity.

## 1. Introduction

### 1.1. Serendipita Indica (S. indica)

Plants and microorganisms coexist in natural ecosystems, which is beneficial to both [1], and due to the fact that plants are constantly exposed to environmental stresses, they strive to improve their resistance to these stresses [2]. One of these methods is the recruitment of beneficial microorganisms that coexist with plants, including bacteria, mycorrhizae, non-mycorrhizae, and fungal endophytes [3]. In return for the valuable services of microorganisms, plants can provide them with 15–20% of their photosynthetic product, which is generally offset by the benefits of microorganisms [4].

An endophytic fungus called *S. indica* is one of the microorganisms that helps plants grow, absorb nutrients, and resist environmental stresses by coexisting with them [5], which shares some features and modes of action with Arbuscular mycorrhiza fungi (AMF) [6]. The fungus, perhaps the most well-known endophyte, was first isolated from *Prosopis juliflora* and *Ziziphus nummularia* in the Tar desert of Rajasthan, India [7]. *S. indica* (formerly known as *Piriformospora indica*) belongs to *Sebacinaceae family* and *Basidiomycota* division [7]; it can grow intercellularly and in-tracellularly and form its pear-shaped chlamydospores in the root cortex of the host plant [8] (Figure 1).

Cell walls are a major barrier to the entry of fungi (beneficial or pathogenic) into plant tissues, and fungi overcome the barrier through the use of cell wall-destroying enzymes (CWDE) and host tissues by destroying plant cell wall components [10]; CWDEs (induced at 5 dpi on dead root material) are a group of enzymes including glycosyl-hydrolases (e.g., cellulases, xylanases, glucanases), esterases, lyases, and oxidoreductases [11,12]. Microorganisms such as *S. indica*, with the activity of CWDEs, destroy plant cell wall polysaccharides and pave the entry pathway to plant cells [11,12]. Also, CWDE production by *S. indica* can be inhibited by glucose or other simple sugars in a metabolic process called glucose repression or catabolism [12,13]

*S. indica* is phylogenetically similar to AMF [12], and it can colonize many host plants and positively affect plant growth and development under environmental stress [14]. As opposed to other root-colonizing microbes, such as Rhizobia, which are species-specific [15], *S. indica* coexists with more than 200 different plant species [16], in-cluding many drought-resistant shrubs and trees, annuals and perennials, angiosperms, bryophytes, pteridophytes, and gymnosperms [4,17]. A major advantage of *S. indica* over AMF is that it can be grown quickly in an artificial medium, making it useful as a biofertilizer in farmlands and gardens [18,19].

Inoculated plants respond positively to *S. indica* by different mechanisms, including widespread growth and root system proliferation [20], early flowering [21], increasing tolerance to abiotic and biotic stresses [22], water and nutrient uptake (phosphorus, nitrogen, potassium, iron, calcium, sodium, and zinc) [23,24], enzymatic and non-enzymatic antioxidant production, phytohormone production (indole acetic acid, abscisic acid, and auxin) [20], photosynthetic activity, seed production [25], and activity of soil microorganisms [26] (Figure 2), some of which will be explained below.

Inoculation of Arabidopsis, Chinese cabbage, rice, and corn roots with *S. indica* causes root proliferation and biomass changes, according to Tsai et al. [27], and it may be an increase in root biomass due to increased indole-3 acetic acid (IAA) production by *S. indica* [28]. Komis et al. found that *S. indica* colonized maize roots after 15 days, which led to their further growth due to stimulating genes involved in microtubule-based processes [29]. Microtubules and actin microfibers play a significant role in root growth by activating cell division, facilitating cell movement and cell wall deposition, and transporting materials within the cells [30]. Numerous studies have shown that Pitef1 gene expression can be used as a marker for plant root colonization by *S. indica* [31,32].

As mentioned, plants provide carbon sources for *S. indica*, which in turn, it improves host plant nutrient uptake [33]. Phosphorus (P) is one of the elements that *S. indica* plays an important role in increasing uptake by the host plant [6]. High P uptake due to plant roots being colonized by *S. indica* reduces malondialdehyde content, increases the concentration of osmolites and nitrogen compounds, maintains the optimal relative water content of leaves, and improves photosystem II efficiency and net photosynthesis rate [34,35,36]; thus, plant colonization by *S. indica*, especially in areas with P deficiency, increases the vegetative growth of plants (the same as *Zea mays*, *Piper nigrum*, *Cicer arietinum*, and *Glycine max*) [6,25]. It should be noted that when *S. indica* is faced with a deficiency of P, a high-affinity P transporter gene (PiPE) is induced, which is responsible for effective P uptake [6].

*S. indica*, in addition to increasing crop production in plants such as rice, barley, black cabbage, and melon [31,37,38], can increase secondary metabolites and phytochemicals in some medicinal and industrial plants [39]. Ghorbani et al. stated that the colonization of tomato roots by *S. indica* increases the accumulation of betaine, glycine, and proline in the roots and the content of photosynthetic pigments [40]. Also, studies have shown that plant root colonization by *S. indica* increased the total amount of piperine and oleoresin in black pepper (*Piper nigrum* L.) [41]; the amount of carvacrol, thymol, and mannol in oregano (*Origanum vulgare*), sage (*Salvia officinalis*) and thyme (*Thymus vulgaris*) [42]; and the amount of chavicol in Basil (*Ocimum basilicum*) [43].

One of the essential roles of *S. indica* in plant coexistence is to help enhance the plant’s resistance to biotic stresses (such as pathogenic fungi and viruses) [3] and abiotic stresses (including cold, heat, salinity, drought, and heavy metal toxicity) [44,45]. One of the most important capabilities of this fungus is that it can adapt and grow under harsh environmental conditions such as lack of access to nutrients, high concentrations of heavy metals, high temperature, and pH level [46]. Mohsenifard et al. also stated that *S. indica* increases plant tolerance to environmental stresses based on plant species’ general and non-specific mechanisms [47].

Research has shown that this fungus helps to improve plant resistance to environmental stresses through enhancing chlorophyll content and photosynthesis rate [48], increasing nutrient uptake and plant growth [49], improving the synthesis of plant antioxidant enzymes and flavonoids [50], immobilizing pollutants [49], clearing ROS, and reducing damage to cell membranes [51]. It has been shown that *S. indica* plays such roles by activating stress-responsive genes in host plants, including dehydration-responsive (DREB2A), pathogenesis (PR), and low-temperature (RD29A) genes [52].

### 1.2. Drought Stress and S. indica

Climate change is one of the most critical issues facing 21st-century humans [53], which mainly impacts water resources and the severity of summer droughts [54]. Thus, climate change has become a significant constraint on crop production, affecting more than 10% of arable land now, and scientists estimate that by 2050, the average crop yields will fall by more than 50% [55]. Drought stress is the most crucial environmental constraint for sustainable agriculture, affecting more than 40% of farms in arid and semi-arid regions [56]. In addition, the growing global population is putting pressure on the food system worldwide. Thus, scientists must look for ways to minimize the adverse effects of drought [57].

As multidimensional stress, drought negatively affects plants’ physiological, morphological, molecular, and biochemical characteristics [58], significantly reducing plant growth, biomass, and yield [59]. Drought stress disrupts cell division, cell enlargement, leaf area, chlorophyll content, transpiration, and photosynthesis [59,60]. Drought can also interfere with the internal uptake and distribution of nutrients, the activities of essential enzymes, and light respiration in plants [53,61]. One of the most important plant approaches in arid areas is stomatal closure and minimization of transpiration [55], which reduces CO_2_ emissions to carboxylation sites and reduces photosynthesis [62]. Decreased CO_2_ fixation under high-light conditions increases light inhibition and, consequently, increases the production of reactive oxygen species (ROS) such as H_2_O_2_ in chloroplasts [63]. Thus, the increase in drought-reactive oxygen species causes oxidative damage to carbohydrates, proteins, lipids, and DNA [64].

Plants have different morphological, physiological, and biochemical responses for their adaptation to drought conditions [65], and in general, these mechanisms can be divided into two categories: stress avoidance and stress tolerance [66]. Plant avoidance responses to drought stress can include leaf morphology and orientation changes, improving water uptake by the wider root system, increasing cuticle resistance, and decreasing stomatal conductance [67]. While osmotic regulation, activation of antioxidant defense systems, and maintaining the cell wall elasticity are three mechanisms that can increase plant toleration to drought stress [68].

The accumulation of low-molecular-weight osmolites (including proline and other amino acids, polyols, and organic acids) is known as osmotic regulation [69], which plays a crucial role in improving plant yield under drought conditions [70]. Plant cells also have enzymatic (such as ascorbate peroxidase (APX), catalase (CAT), and superoxide dismutase (SOD)) and non-enzymatic (such as glutathione, ascorbic acid, and carotenoids) antioxidant mechanisms to protect themselves against drought-induced ROS [27,71]. In addition, abscisic acid (ABA) plays a significant role in responding to drought in plants, and many genes respond to this stress through ABA-dependent pathways [72]. Also, hormones such as auxin, ethylene, cytokinin, gibberellin (GA), jasmonic acid (JA), and brassinosteroids increase plant drought tolerance [73]. On the other hand, plant cells have several mechanisms for dealing with high mechanical impedance in soil, including increasing the shedding of border-like cells, changing the orientation of cellulose microfibril deposition, producing ethylene, and increasing root cap secretory cells [74].

Plants are also controlled by a chain of molecular networks against drought stress [75]. In general, all genes and proteins involved in plant drought resistance are divided into several groups, including signaling and transcription factors, active osmotic compounds, and agent proteins [75,76]. One of the most important defense mechanisms that plants use to cope with drought is miRNA regulation [77]; miRNAs, which are small 21-nucleotide RNAs, play a key role in regulating gene expression by inhibiting or degrading mRNA translation [47]. Recent studies have shown that the expression of miRNA854a, miRNA474 [78], miRNA166, miRNA167, miRNA319, miRNA413 [79], miRNA18, miRNA71 [80], and miRNA159 [47] increased under drought stress conditions in rice, with miRNA319 and miRNA166 controlled by GA and miRNA413 and miRNA167 by ABA [79].

Furthermore, Hundertmark and Hincha reported that LEAs (late embryogenesis abundant) expression in different plants increases their resistance to drought [81]. These proteins consist of seven different groups, each with its own characteristics [82]. Studies have shown that these functional proteins accumulate in plants in response to stress and increase plant resistance [36]. Magwanga et al. stated that LEA2 gene expression in transgenic Arabidopsis improves root length and plant drought resistance [83]. Azizi et al. also reported that the expression of the Dehydrin TAS14 gene, one of the most important groups of LEA, showed a significant increase in tomato yield under drought conditions [36].

In recent years, scientists have sought appropriate, inexpensive, and practical solutions to decrease the adverse effects of drought stress on crops [84]. There has been a high focus on crop breeding to provide drought-resistant varieties in past decades, but genetic advances in this area have generally been slow due to the multi-gene nature of the tolerance process [85]. Therefore, scientists have focused on the positive effect of soil microorganisms and their coexistence with crops, where these microorganisms can play a crucial role in reducing the effects of drought and consequently increase plant resistance [84,86]. One of these beneficial microorganisms is *S. indica*, which can colonize the roots of a wide range of plants and increase their drought resistance by affecting the physiological characteristics of the host plant [87]. Various studies have shown the effect of *S. indica* in reducing the adverse effects of drought on the following plants: *Brassica rapa*, *Hordeum vulgare* [25], *Solanum melongena* L. [88], Medicago, rice, arabidopsis, and maize [27].

In summary, the positive role of *S. indica* in decreasing drought effects in plants can be divided into four main groups: (1) improving water and nutrient uptake [55,88], (2) increasing chlorophyll synthesis, (3) accumulation of high proline content, and (4) protection against ROS by overproduction of antioxidant enzymes [88]. The role of *S. indica* in increasing the growth of host plants and helping to better absorb nutrients such as N and P as well as the effect on better water uptake by plants, is known; scientists attribute the positive effect of *S. indica* to improving plant yield under drought conditions to the origin of this fungus, which is from arid areas [55]. Research has shown that drought severely suppresses key enzymes in nitrogen metabolisms, such as glutamate synthase, glutamine synthetase, and nitrate reductase [89]. However, this suppressive effect was not observed in plants whose roots were colonized by *S. indica*, and it was shown that the fungus helps improve N uptake by plants under drought conditions [89].

Ghaffari et al. also stated that *S. indica* improves the redistribution of resources in the host plant and protects them from the negative impact of drought [85]. In addition, it has been reported that plant colonization by *S. indica* expands their root systems [70]; therefore, it improves water and nutrient uptake, which ultimately increases plant biomass, modulates leaf temperature, reduces leaf wilting, and improves stomatal closure [27]. There is also evidence that *S. indica* can decrease drought-induced senescence by controlling autophagy [85]; Ghaffari et al. clearly showed that the expression of the EXO70B1 gene regulates autophagy in plants is significantly increased when plant roots are colonized by this endophytic fungus [85]. Sun et al. identified the effect of *S. indica* on the expression of the stomatal regulator of Chinese cabbage leaves (Calcium signaling), where it causes pores to close under drought conditions [90].

Other positive effects of *S. indica* in decreasing drought stress risks include preventing the degradation of thylakoid and chlorophyll proteins and reducing photosynthetic efficiency [51,90]. Blanco et al. reported that colonization of plant roots by *S. indica* regulates photosynthesis-related proteins under drought stress conditions, including components of LHC-I and LHC-II photosystems, key enzyme optical respiration, ferrodoxin, phosphoglycolate phosphatase, and photosystem complex proteins (PSBO, PSAG, and PSAK) [91]. 

Recent studies have shown that inoculation of plants with *S. indica* under drought stress increases proline content and decreases malondialdehyde (MDA) accumulation [27,87]. In addition, *S. indica* regulates the expression of the P5CS gene, which increases proline accumulation [92]; this endophytic fungus also regulates the expression of genes related to total antioxidant capacity (TAC), which increases the plant’s resistance to drought [51]. Research has also shown that *S. indica* increases SOD, CAT, and APX and regulates their related genes in plants [27,73]. As mentioned, ABA plays a significant role in increasing plant resistance to drought stress [87,93], and Mohsenifard et al. found that the expression of some miRNAs (such as miR396 and miR159), which are involved in the regulation of ABA, are increased by inoculation with *S. indica* [47].

Factors that affect the resistance of plants to drought are the axial growth pressure of roots (σ_max_) [94]; its amount in different species and genotypes of plants is in the range of 200 to 1300 kPa [95]. σ_max_ is controlled by internal root factors (including cell wall expandability, osmotic potential, and cell wall pressure) and external factors (such as soil temperature and matrix potential) [94,96]. Studies on *Triticum aestivum* L. and *Zea mays* L. have shown that colonizing plant roots with *S. indica* increases σ_max_ and thus increases the length and volume of plant roots under drought conditions, which ultimately improves water and nutrient uptake [20,97].

According to recent studies, simultaneous inoculation with *S. indica* and AMF increases host plant biomass and reduces drought stress. [98,99]. The researchers also showed that simultaneous inoculation with *S. indica* and AMF could significantly increase chlorophyll content, proline content, and relative water content in *Eleusine coracana* under drought conditions [86]. Heidarianpour et al. also stated that simultaneous inoculation with these two fungi improved the absorption of Ca, P, K, and Na in tomatoes under drought stress [98]. However, it should be noted that water deficiency reduces root colonization by fungi; scientists primarily attribute this to the AMF’s urgent need for water for growth and, secondly, the reduction in carbohydrate supply by the host plant [86,99].

### 1.3. Salinity Stress and S. indica

Salinity is one of the biggest environmental threats, especially in arid and semi-arid regions [19], significantly affecting the health of soil, water, and crops [100]. According to research, 20% of irrigated land is currently affected by this stress [40], and it is predicted that by 2050, almost half of agricultural land will be affected by salinity stress [101]. Increased salinity in agricultural soils is due to poor soil management, limited rainfall, high temperatures, high evapotranspiration, and inappropriate agricultural practices [102] that strongly impact agricultural production [103]. Therefore, finding solutions to reduce the negative effects of this stress can significantly improve agricultural productivity [104].

Saline soils contain different anions and cations, including Na_2_CO_3_, CaSO_4_, Na_2_SO_4_, MgSO_4_, MgCl_2_, and KCl [8], in which NaCl is predominant [105]. In general, high concentrations of sodium (Na^+^) alter the complete structure of the soil by reducing soil porosity, soil aeration, and water conduction, which ultimately reduces the soil water potential and prevents the uptake of water and minerals by plants [102]. Sharma et al. also reported that salinity stress impairs the uptake of water and minerals by plants, thereby reducing their growth and biomass [106]. According to reports, increased soil salinity has disturbed plants’ ionic and osmotic balance, which is the greatest danger to plants and can lead to their death [105]; disturbing this balance causes physiological and metabolic changes [107], which will be discussed below.

Among the dangers of salt accumulation in the rhizosphere for plants is severe damage to the roots, which reduces plants’ uptake of water and nutrients [19]. Salinity stress prevents adequate water uptake by reducing the osmotic potential of the soil [108]; thus, it causes physiological drought of the plant, which ultimately affects plant growth and reduces its yield [19,109]. Ion toxicity, also due to excessive accumulation of Na^+^ and chloride (Cl^−^), reduces the absorption of many nutrients, including nitrogen (N), potassium (K), calcium (Ca), and phosphorus (P) [110]. Plants normally have 1–10 mM Na^+^ and 60–100 mM K^+^ in their cytosol [111]; increased Na^+^ contents, due to its competitive nature against K^+^ in binding to necessary sites, causes impaired K^+^ absorption by roots and consequently excessive depletion in different organs of plants [110,112].

According to researchers, salinity stress causes hyperosmotic and hypertonic stresses in plants, which closes stomata, prevents leaf expansion, and increases the accumulation of toxic ions in the shoots of plants [102]. High soil salt concentrations also cause oxidative stress through the production of ROSs [113], including superoxide ions (O_2_^−^), hydroxyl radicals (OH), and hydrogen peroxide (H_2_O_2_) [114], which adversely affect the function and natural structure of cellular components such as nucleic acids, lipids, and proteins [115]. Salinity leads to the cessation of plant growth by reducing photosynthetic CO_2_ uptake [116]. Salinity stress can also affect photosynthesis through chlorophyll peroxidation, dysfunction of pigment–protein complexes (PPCs), and degradation of enzymes involved in photosynthesis, effectively reducing biomass and yield [8,117].

Plants have different mechanisms to deal with the harmful effects of salinity, and in general, these reactions can be divided into biochemical and physiological [118]. Under salinity stress, plants can survive by limiting Na^+^ uptake, reducing Na^+^ transfer from roots to shoots, and removing Na^+^ from root cells [8]. To prevent the increased Na^+^ toxicity in shoots, plants can prevent Na^+^ delivery to photosynthetic tissues and do so in two ways: (1) by controlling the loading of Na^+^ in the xylem and (2) by increasing the rate of Na^+^ re-transfer through phloem [110]. For this purpose, plants have genetic mechanisms that help increase their resistance by limiting the entry of sodium into the plant by selectively absorbing ions or increasing the flow of sodium to the aplastic spaces and culture medium [119].

The results of studies have shown that one of the strategies that plants use to protect themselves under salinity stress conditions is to increase the level of ABA in the shoot, which leads to the closure of the stomata, which reduces plant growth and decreases consumption of water [93,110]. At the same time, increasing the level of ABA in the roots of plants controls the loading of Na^+^ and K^+^ in the xylem and adjusts their delivery to the shoots to achieve osmotic regulation [93,120]. Yun et al. reported that increasing access to K^+^ improves stomatal performance and thus plant efficiency under salinity stress, and preserves a set of ATPs involved in synthesizing organic osmolytes [110].

Another defense mechanism of plants to counteract excess sodium accumulation and potassium deficiency is to regulate the expression of several membrane transport genes responsible for the uptake, segmentation, or displacement of Na^+^ and K^+^ [121], including the NHX and salt overly sensitive (SOS), which are involved in Na^+^ isolation in vacuoles and Na^+^ excretion from cells [121,122]. NHXs play an important role in ionic homeostasis by isolating Na^+^ and activating K^+^ uptake in vacuoles [123], maintaining intracellular pH and regulating cellular turgid and stomatal function [123,124]. Researchers have shown that tomato contains four isoforms [125], corn and rice contain six isoforms, and Arabidopsis contains eight isoforms of NHX [101]. Wu et al. reported mutations in SOS family genes in plants under salinity stress [126]. This family of genes is involved in inhibiting Na^+^ from reaching photosynthetic tissues [113], and since these genes are antiporters of Na^+^/H^+^, they also play a role in the transfer of Na^+^ to apoplastic spaces [113,127]. Also, higher expression of SOS genes in the tissues around the xylem indicates their role in redistributing Na^+^ between roots and shoots [113].

As mentioned, the growth of salt accumulation in the rhizosphere of plants increases ROS [128], and among the protective mechanisms that plants use to reduce the adverse effects of salinity and ROS is the increased activity of enzymatic (such as SOD, GR, APX, guaiacol peroxidase (POD), and CAT) and non-enzymatic antioxidants (such as glutathione (GSH), ascorbic acid (AsA), phenols, tocopherols, and thiols) [129,130,131], hormones, and secondary metabolites [129]. In fact, the combination of enzymatic and non-enzymatic antioxidants dramatically increases the tolerance of plants to salinity stress [132]. 

The production of several compounds such as glycine, betaine, and proline constitutes other physiological mechanisms of plants in response to salinity and for improving osmotic regulation [113]. Proline accumulation helps plants stabilize cell membranes and regulate usable nitrogen [128], and it should be noted that measuring the proline content of plant organs is an effective way to determine salinity stress and its effect on plants [133]. Aquaporins (including TIPs and PIPs) also significantly increase plant resistance to salinity [122]; These proteins, abundant in tonoplasts and plant cell plasma, play an important role in intercellular and intracellular water transport [122,134].

In recent decades, various methods for saline soil remediation have been developed, including chemical remediation and freshwater irrigation, which are not widely used due to high costs and relatively minor effects [135]. Several attempts have also been made for genetic modification of salinity-resistant crops, but they have been unsuccessful due to the involvement of multiple genes and the multistage biochemical pathway [135,136]. Hence, scientists have had to find cost-effective and practical solutions to increase plant resistance in salted environments [135,137]. One of the methods that scientists have considered is using natural soil microorganisms to reduce the adverse effects of salinity on plants [136], due to its inexpensiveness, fast yield, and environmental friendliness [138].

*S. indica* is one of the soil microorganisms that has been shown to increase plant resistance to salinity stress [139]. These plants include rice, barley, wheat, tomato, Medicago truncatula, and Arabidopsis [14,140]. *S. indica*, in addition to causing morphological and physiological changes in plants under salinity stress, causes major changes in various molecular mechanisms to increase the resistance of plants to salinity [8] (Figure 3).

Researchers have shown that colonization of roots by *S. indica* increases biomass, root and shoot branching, root and shoot length, and fresh and dry weight of plants under salinity stress [40,101,106]. Ghorbani et al. stated that tomato inoculation by *S. indica* improves plant yield under saline conditions via improving plant water uptake [40], due to the regulation of aquaporin gene expression [141]. Research has also shown that with increasing soil salinity, the use of *S. indica* increases the uptake of 14 metabolites and ions by plants [18], including N, Ca, K, and P [122]. It should be noted that *S. indica* reduces Na^+^ accumulation in plant shoots and roots, and has no significant effect on Cl^−^ accumulation in plants [122]. 

**Figure 3 biology-11-00952-f003:**
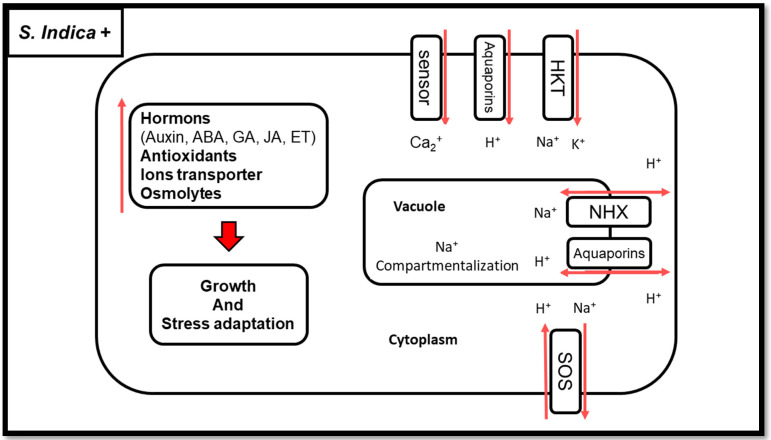
A schematic representation of root cells colonized by *S. indica* under salinity conditions. Nomenclature is as proposed by Gazara et al. [142]. HKT—high K^+^ transporters; SOS—salt overly sensitive; NHX—sodium/hydrogen antiporter; Na^+^—sodium; H^+^—hydrogen; K^+^—potassium; ABA—abscisic acid; GA—gibberellin; JA—jasmonic acid; ET—ethylene.

Another positive effect of *S. indica* on plants under salinity stress is the improvement of the photosynthetic state [40], including increasing photosynthetic pigments [40,140], total chlorophyll content (chlorophyll a and b) [101,106], and the photochemical efficiency of photosystem II [40]. However, it is still unclear how *S. indica* influences plant growth under salinity [14]. Previous research has shown that *S. indica* improves the osmotic state of plants; however, the effect of this endophytic fungus on the readjustment of ionic imbalance in plants is unknown [14,18]. 

As mentioned, Na^+^ and K^+^ compete for binding to plant uptake sites; however, Yun et al. have shown that *S. indica* increases K^+^ uptake and prevents Na^+^ uptake by plants in saline soil [110]. Also, Hassani et al. have confirmed the above results in their research on melons [8]. Research on *S. indica* has shown that ENA ATPase plays an essential role in increasing the tolerance of this endophytic fungus under salinity stress (it should be noted that *S. indica* can grow even in a high salt concentration of 219.14 mM), and functional analysis has shown that these ATPases extrude Na^+^ and K^+^ [14,143]. Yun et al. stated that roots and shoots of *Zea mays* L. inoculated with *S. indica* contained more K^+^ under salinity, possibly due to improved stomatal conductance and reduced K^+^ flow from the roots [110]. The most important ways of loading K^+^ are HKT (high K^+^ transporters), HAK (high-affinity K^+^ transporter), and SKOR (outward rectifying Shaker-like K^+^ channel), but which of them is most affected by *S. indica* has not been studied yet [110].

Abdelaziz et al. found that a rise in K^+^ concentration in Arabidopsis inoculated with *S. indica* was associated with the increased expression of KAT1 and KAT2 ion channels [144]. Since ABA is considered a stress hormone, research has shown that it can increase K^+^ accumulation in plant roots under salinity stress [110]. Research on Arabidopsis has shown that ABA levels are increased in roots of plants inoculated with *S. indica*, so it can be concluded that increased ABA induced by *S. indica* can affect the host’s initial defense against salinity [93]. However, the molecular pathways of these processes are still unclear and need further study [110].

Additionally, *S. indica* increases salinity tolerance by altering the level of enzymes that eliminate ROS from plants [3,101]. Root colonization by *S. indica* significantly increases antioxidant and non-antioxidant enzymes such as POD, SOD, CAT, CAT, APX, GR, dehydroascorbate reductase (DHAR), monodehydroascorbate reductase (MDHAR), and GSH in roots and shoots of plants exposed to salinity [5,101]; at the same time, it reduces H_2_O_2_ and MDA in various plant organs [5]. Furthermore, inoculation of plants with *S. indica* increases the levels of plant hormones such as salicylic acid (SA), gibberellic acid (GA), betaine, glycine, and proline, which improves the plant’s resistance to the harmful effects of environmental stresses, especially salinity [8,40,113]. Sharma et al. have also reported that *S. indica* increases the aloin, flavonol, flavonoid, and phenol content of plants, reducing oxidative stress caused by salinity [106].

Another mechanism by which *S. indica* improves plant resistance to salinity stress is to help with the rapid accumulation of chloroplast calcium sensor proteins (CAS), which regulate Ca_2_^+^ agglomeration in leaves and control the aperture of leaf stomata [145]. Numerous studies have shown that *S. indica* helps plants overcome salinity by increasing SOS and NHX genes’ family expression [101,113,122]. Ghorbani et al., in their study on tomato seedlings under salinity stress, found that *S. indica* increased its resistance by regulating the expression of SOS1 and NHXs in plant roots [122]. In their studies on rice, Porcel et al. showed that the coexistence of *S. indica* with plant roots increases the expression of OsSOS1 and OsNHX3 genes under salinity stress [108]. *S. indica* has also been reported to help increase plant salinity resistance by increasing the expression of TIP and PIP family proteins, which regulate water status, nutrients, and ionic homeostasis [122].

## 2. Conclusions

Due to *S. India*’s coexistence with many plants, ease of culturability compared to AMF, and positive morphophysiological and molecular effects on plants, it can be widely used as a biofertilizer. However, extensive studies are still needed to understand how this large-scale endophytic fungus functions. As reported in this study, this fungus has a positive effect on raising plants’ resistance to stresses such as drought and salinity, so it can be used in modern agriculture to address the increased risk of environmental stresses. Nevertheless, parts of how the fungus functions under environmental stresses, especially salinity stress, remain unknown. Thus, it is suggested that future research focus on the plant’s molecular mechanisms for a better understanding of how *S. indica* works; Knowing more about this topic will help us determine whether this fungus can enhance agricultural production under stresses.

## Figures and Tables

**Figure 1 biology-11-00952-f001:**
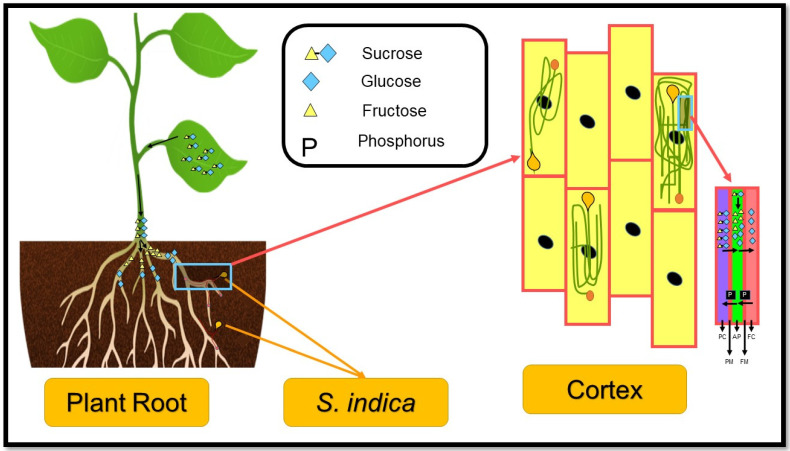
A schematic of *S. indica* chlamydospores growth in the plant’s root cortex. Nomenclature is as proposed by Opitz et al. [9]. PC—plant cytosol; PM—plant plasma membrane; AP—apoplast; FM—fungal membrane; FC—fungal cytosol.

**Figure 2 biology-11-00952-f002:**
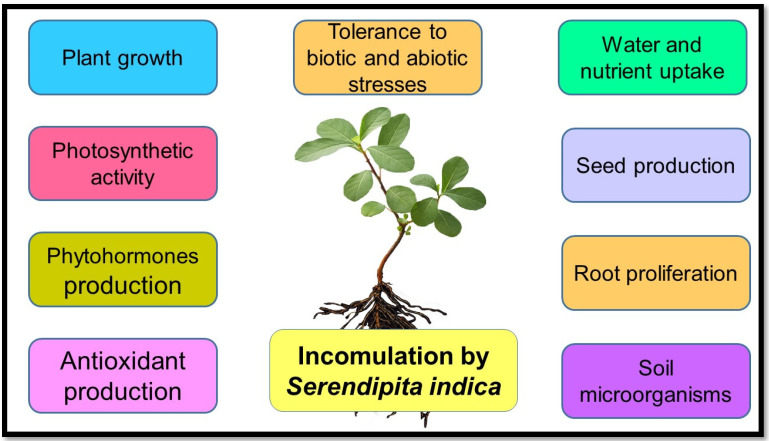
Positive morphological, physiological, and molecular effects of inoculation of plant roots with *S. indica*.

## Data Availability

Not applicable.

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
