# Peer review of "The Role of Serendipita indica (Piriformospora indica) in Improving Plant Resistance to Drought and Salinity Stresses"

_biology, 2022, doi:10.3390/biology11070952_

Round 1

Reviewer 1 Report

NA

Author Response

Dear Sir / Madam

The manuscript was rechecked, and grammar was modified as much as possible.

Thanks again for all your helpful comments

Best regards

M. Boorboori

Reviewer 2 Report

The Authors partially followed the comments contained in the review.

The manuscript by Boorboori and Zhang  "The role of Serendipita indica (Piriformospora indica) in improving plant resistance to drought and salinity stresse" in this form can be accepted after the correction of the scheme of root colonization pattern by S. indica (Fig. 1) and sentences about CWDE (P. 2 L.81).

Diagram taken from Opitz et al. (2021) paper was used in a very shortened form, which meant that the scheme contains a serious error and cannot be used in this form. The authors presented only intracellular hyphae, and did not show that in the cortex layer of plant roots there are hyphae in the intercellular spaces, through which the hyphae move and the infection of cells occurs from these spaces. The authors mention the presence of two types of S. indica hyphae in the root cortex in the sentence preceding the scheme (P 2, L. 49) "can grow intercellularly and intracellularly and form its pear-shaped chlamydospores”, but do not indicate it in the figure. First of all, if the authors want to use the scheme taken from the publication of Opitz et al. (2021) it must be emphasized that S. indica, in interaction with the plant, obtains only glucose and not fructose, which comes from cleavege of plant sucrose, in which fungal invertase participates. On the other hand, the plant obtains inorganic phosphorus in this interaction, and elicitors are transferred from the S. indica hyphae to the plant, which shape the plant's immune / defense response to infection.

The schema (Fig. 1) must be absolutely corrected.

 The sentence in which the Authors give examples of

CWDE P. 2 L.81 should also be corrected.

Examples of hydrolases which are the main CWDEs should be given, remove the word “glycosyl” and enter glycosidases or glycosyl-hydrolases

Change: "enzymes including hydrolase, esterases, glycosyl, lyases, and oxidoreductases"

On "enzymes including glycosyl-hydrolases (eg. cellulases, xylanases, glucanases), esterases, lyases, and oxidoreductases".

 Fig. 3 - in the title of Fig. 3 abbreviations should be added.

Author Response

Dear Sir / Madam

Thanks for your helpful comments, you can find modification detail below:

  • The manuscript was rechecked, and grammar was modified as much as possible.
  • Figure 1 was redesigned completely.
  • About CWDE part, I followed your suggestions.
  • 3 abbreviations were added.

Thanks again for all your helpful comments

Best regards

M. Boorboori

Reviewer 3 Report

Review

The role of Serendipita indica (Piriformospora indica) in improving plant resistance to drought and salinity stresses

The authors give a nice overview about the mechanisms through which, Serendipita indica, an endophytic fungus with a broad range of hosts, alleviate effects of drought and salinity stress in plants. The review starts with a general description of previous research on S. indica, and then it focuses specifically on work done with the fungus in relation to the two environmental stresses (drought and salinity). The authors cover the topic adequately and provide enough references. However the language is problematic in some parts, making it difficult to follow the narrative.

In detail, these are a few points that need to be revised:

·        Line 13: to use them to reduce

Change to “to use them in order to reduce”

·        Line 15: Moreover, this study has investigated the impact of these stresses on plants and the solutions plants have to deal with these stresses;

Please rephrase. The second part of this sentence does not make sense.

·        Line 18: Plant stresses are

Change to “Plant stresses is”

·        Line 20: prevents

Change to “will prevent”

·        Line 38-39: One of these methods is absorbed beneficial microorganisms that need foods to survive, including bacteria, mycorrhizae, non-mycorrhizae and fungal endophytes.

Please rephrase to make sense. E.g. instead of absorbed use the recruitment of

·        Line 42: generally compensated by microorganisms' benefits

Please rephrase

·        Line 45: its features and how it works are very similar to

Change to “shares some features and mode of action with”

·        Line 79: through cell wall-destroying

Change to “through the use of cell wall-destroying”

·        Line 80: and find a way to enter

Change to “and”

·        Line 81: induced at five dpi on dead root material

In which plant and what is the experimental setup? Or this happens in every plant material it colonizes?

·        Line 87: S. indica is phylogenetically similar to AMF

Please provide references for this statement

·        Line 98: toleration

Change to “tolerance”

·        Line 98-101

Please state what is the effect of colonization and not just e.g. water and nutrient uptake but improved or increased water and nutrient uptake.

·        Figure 2: Incomulation

Change to “inoculation”

·        Figure 2

The legend of the figure is not explaining what is showing on the figure, which itself is a bit vague and confusing. Please make both the figure and the legend more informative otherwise exclude them completely.

·        Line 139: these fungi

Changes to “the fungus”

·        Line 141-144

This is something studied in Phoebe zhennan and it happens under drought conditions. It is also not related to S. indica colonization. If it is the authors’ interpretation of what it might happen when S.indica is present, please rephrase so it is clearer to the reader.

·        Line 163: temperature, and pH levels

Please indicate what kind of temperature and pH levels. Extreme? High? Low?

·        Line 202:toleration

Change to “tolerance”

·        Line 228: LEAs

Please explain what LEA stands for. The same should be done for any other acronym introduced for the first time in the text.

·        Line 386-389

Please rephrase to make sense

·        Line 399

The word intracellular is used twice. Maybe intercellular?

·        Figure 3

Figure 3 is also not explained adequately. Please add a few sentences to describe the mechanisms.

·        Line 467

What HKT, HAK, SKOR stand for? Maybe write it on the legend of figure 3

·        Line 504: on how this large-scale

Change to “to understand how this large-scale”

·        Line 509-511

Please rephrase to make sense

·        References

Please revise the references since some of them are incomplete (e.g. 17, 41, 63, 69, 95, etc). In addition, the Latin names of the organisms have to be presented in italics and some words are in capital letters for no obvious reason.

Author Response

Dear Sir / Madam

Thanks a lot for your helpful comments. You can find all the change details below:

  • Line 13: It changed to “to use them in order to reduce”
  • Line 15: It was rewritten.
  • Line 18: It changed to “Plant stress is”
  • Line 20: It changed to “will prevent”
  • Line 38-39: It was rewritten
  • Line 42: It changed to “is generally offset by the benefits of microorganisms”
  • Line 45: It changed to “shares some features and mode of action with”
  • Line 79: It changed to “through the use of cell wall-destroying”
  • Line 80: It changed to “and”
  • Line 81: according to the reference, 5 dpi is for all plants.
  • Line 87: reference was added
  • Line 98: It changed to “tolerance”
  • Line 98-101: all the details about the effect of colonization are explained after figure 2 with details.
  • Figure 2: It changed to “inoculation”
  • Figure 2 summarizes the effect of plant root colonization by fungi, which is summarized, but the explanation of each of these sections is given in the following paragraphs, which I hope is satisfactory.
  • Line 139: It changed to “the fungus”
  • Line 141-144: two more references were added, and the sentence was rewritten to show Indica effect of increasing P uptake and how it affects drought stress.
  • Line 163: I already added high concentrations of heavy metals, temperature, and pH level.
  • Line 202: It changed to “tolerance”
  • Line 228: LEAs (late embryogenesis abundant) was added.
  • Line 386-389 was rewritten and added to the previous sentence.
  • Line 399: It changed to “intercellular”
  • Figure 3 is the summary of all salinity parts, and all the parts of the figure were explained with detail in that part. If the dear reviewer atill wants to add more details on the figure I can do it but I think it will be the repetition of the text.
  • HKT, HAK, SKOR were explained in the text.
  • Line 504: It changed to “to understand how this large-scale”
  • Line 509-511: It was rewritten
  • All the References were rechecked and modified as much as possible.

Thanks again for all your helpful comments

Best regards

M. Boorboori

This manuscript is a resubmission of an earlier submission. The following is a list of the peer review reports and author responses from that submission.

Round 1

Reviewer 1 Report

PGPR and Stress components has become one of the main research for the last 2 decades. The current article is not new. Authors chosen the organism is new to the PGPR world at least for the authors reading first time.
English language is not standard, could be improved. 
Lack of novelty and new topic.

Reviewer 2 Report

The work concerns an important and current topic of counteracting drought and salinity stress with the use of endophytic fungi.

The importance of endophytic microorganisms should be described in more detail at the beginning of the introduction.

It is necessary to explain when and why the generic name Piriformospora was changed to Serendipita. Plant species that can host S. indica and species other than S. indica belonging to the genus Serendipita should be mentioned. Please explain the geographic distribution of this species. Can they colonize organisms other than plants? Can they be harmful or affect plants or be their pathogens?

It is necessary to organize the mechanisms of Serendipit's activity in conditions of abiotic stress, drought and salinity, as well as biotic stress (infection by phytopathogens): e.g. Author mention the word "osmolytes" on line 108 and they give the definition of osmolytes on line 170 and do not write there but "betaine, glycine" which they previously mentioned on line 118The authors omitted a very important aspect of S. indica interaction with bacterial plant symbiontsand a great number of known examples of the formation of S. indica consortia with strains of endophytic bacteria that directly affect plant growth and by limiting the development of phytopathogens, mainly pathogenic fungi, e.g. belonging to the genera Fusarium and Rhizoctonia.The scheme (Fig. 1) is too general and requires extensive supplementation to include the different mechanisms of S. indica interaction. It should be noted how inoculation and colonization by an endophytic fungus affect individual properties / activities - it should be clearly stated whether they are strengthened or weakened. It is not clear what "Soil microorganisms" means. S. indica is characterized by the ability to produce enzyme complexes that strongly affect / degrade the cell walls of plants and microorganisms - the manuscript lacks information about CWDE (cell wall degrading enzymes) produced by S. indica. Detailed comments: Title - Change Serendipita indica (Piriformospora indica) on Serendipita (Piriformospora) indica,

Line 36 abbreviation (S. indica) in the title of subchapter 1.1. is redundant Line 61 and Fig. 2 S. Indica - typing error - change to lower case letter in species name - S. indica

Line 107 S.indica - no space between the abbreviation of the generic name and the species name.

Reviewer 3 Report

The paper discusses an important but little highlighted fungus that improves plant stress tolerance.
Editing is required to help understand several sentences

I did not feel that the text covered everything that was implied in the abstract

How do different isolates differ? 

What for me was missing was where does this endophytic fungus travel in planta     what is its means of ingress into the plant   what are the responses of the fungus alone to salinity stress

Does colonization harm plant growth  ie trade off for protection v colonization?

liked  background general information but looking for more facts as the manuscript goes on

the ROS connections are expected now  not so novel   could be reduced in amount of discussion    also all the proline discussions etc 

sections could be smoothed out   eg role of miRNAs 
